# A Study on a Matching Algorithm for Urban Underground Pipelines

**Shuai Wang [1], Qingsheng Guo [1,2,*], Xinglin Xu [1] and Yuwu Xie [1]**

[1]   School of Resource and Environmental Science, Wuhan University, Wuhan 430072, China
[2]   State Key Laboratory of Information Engineering in Surveying, Mapping and Remote Sensing,
     Wuhan University, Wuhan 430072, China
*   Correspondence: guoqingsheng@whu.edu.cn; Tel.: +86-138-7144-1265

**Abstract:** Urban underground pipelines are known as "urban blood vessels". To detect changes in integrated pipelines and professional pipelines, the matching of same-name spatial objects is critical. Existing algorithms used for vector network matching were analyzed to develop an improved matching algorithm that can adapt to underground pipeline networks. Our algorithm improves the holistic matching of pipeline strokes, and also a partial matching algorithm is provided. In this study, appropriate geometric measures were selected to calculate the geometric similarity between pipeline strokes in their holistic matching. Existing methods for evaluating similarities in spatial scene structures in partial underground pipeline networks were improved. A method of partial matching of strokes was additionally investigated, and it compensates for the deficiencies of holistic stroke matching. Experiments showed that the matching performance was good, and the operation efficiency was high.

**Keywords:** matching; partial stroke matching; underground pipelines; stroke

## 1. Introduction

Urban underground pipelines, which are known as "urban blood vessels", consist of a series of pipelines for water supply, drainage, gas, heat, electricity, communications, television transmission, and other industries [1]. Underground pipeline data play a key role in smart cities, disaster evaluation, and so on [2–4]. For different applications, there are differences in the data model, location accuracy, and attribute information, especially between integrated underground pipeline data and professional underground pipeline data, which are collected by different departments. Integrated underground pipeline data are accurate and general, while professional underground pipeline data express and contain detailed attribute information [5]. As a result, there are two sets of data and two systems. This management model has seriously affected the integration and sharing of pipeline data between different departments. The intention of this work is to achieve a matching algorithm for urban underground pipelines that can be widely applied and used to promote data sharing.

Previous studies on vector data matching have provided sound ideas and useful methods for urban underground pipeline matching [6–11]. However, the spatial data are usually organized by the traditional "node–segment" method, which has some obvious defects. First, it is difficult to analyze and apply the characteristics of linear elements as a whole. Second, it is not easy to express the change and distribution of geographical phenomena along the linear elements. Therefore, the spatial data organization of urban underground pipelines using the "node–segment" approach excludes several discrete segments from the set of basic matching units, even though they represent the same real-world pipeline. Thus, it is difficult to match integrated pipelines and professional pipelines. For these reasons, the "good continuity" principle, proposed by Gestalt, is often used as a guide to establish

geometric connection rules [5,12]. This principle was combined with the characteristics of road and water network spatial data to propose the stroke concept [12,13]. From a local indicator, two road segments can be connected on the condition that their deflection angle is less than a specified threshold. The deflection angle, which ranges from 0° to 180° represents the degree of deviation formed by two linked road segments [14], as shown in Figure 1. Figure 1a shows the pipeline spatial data organized in the traditional "node–segment" manner, and Figure 1b shows the data processed by stroke connection. There are four strokes: stroke1<a, b, c, d>, stroke2<e, f>, stroke3<g>, and stroke4<h>. "Good continuation" usually refers to an approximately level angle (180°) between segments. The geometric morphology of a stroke is characterized by a smooth and continuous distribution in space, which maintains the integrity of urban underground pipelines. Therefore, the stroke can be used as the basic unit for matching and analyzing urban underground pipelines.

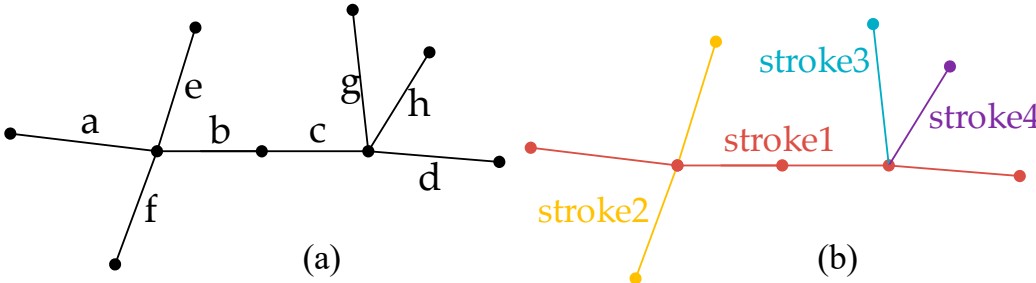

**Figure 1.** (**a**) Segments of pipelines; (**b**) strokes of pipelines.

The matching algorithm proposed in this paper applies the stroke concept. The production of a stroke from discrete segments, includes the principles of attribute information and included angle coherence. If the urban underground pipeline data contain inconsistent or are missing attribute information (for example, an integrated pipeline that only retains the basic information of the pipeline, such as the spatial location, pipe diameter, material, and burial depth), then the main and branch pipes are not recorded. In the absence of the recorded correlation relationship, the principle of angle coherence is adopted in this paper to process the connection of the pipeline segment into a stroke.

In the process of matching, measures of similarity are particularly important. Xavier et al. organized matching measures according to the nature of the measured quantity: geometry, topology, attributes, context, and semantics. Geometric measures refer to the locations of features, as well as their area overlap, geometric properties (size, area, etc.), and shape (e.g., elongation). Topological measures assess the relationships between two nearby features. Attribute measures evaluate the non-geometric properties of a spatial object (e.g., name), and context measures reveal the geographic context of a feature relative to its neighborhood. Finally, semantic measures compare the similarity between different concepts [15]. For network matching, Walter and Fritsch proposed using the topological relation connection as part of their matching method. The connected approach counts the number of features that are connected to another one [16]. Olteanu-Raimond and Mustière proposed the neighborhood criterion as a topological measure. This criterion is based on the assumption that if two edges are similar, then their neighbors should also be similar [17]. Underground pipelines form an organic network as a whole. Each stroke (each basic matching unit) in the network has a certain topological connection with its neighboring strokes, and thus, is not an independent entity.

In the data of underground pipelines with different current situations, an entity often varies as a result of human activities or social development. There are two specific forms of variation: morphological differences and topological differences. A morphological difference mainly manifests as a change in the individual itself, including pipeline growth, shortening, appearance, and disappearance. Differences in the topological structure caused by the present situation mainly manifest as pipeline splitting, merging, and structural changes. In this case, the topological relationship between pipelines changes because of the rectification of underground pipelines. In the face of performance differences

in the above-mentioned underground pipelines, it is impossible to implement the entire matching algorithm by using strokes, because the topological structure of underground pipelines is inconsistent, so the stroke cannot find a matching object in geometric form.

This paper focuses on the geometry matching of underground pipeline data, and the goal is to select appropriate geometric measures for calculating the geometric similarity between pipeline strokes. Existing methods for evaluating similarities in spatial scene structures in partial underground pipeline networks are improved in this work. By calculating the geometric similarity and structural similarity of the network (neighborhood space) where a stroke is located, the matching results are more reliable. For morphological differences and topological differences in pipeline strokes, a method for the partial matching of strokes was additionally investigated, and it compensates for the deficiency of holistic stroke matching. In both the holistic and partial data matching, stroke-based methods are applied.

The remainder of this paper is organized as follows. Section 2 introduces some previous vector matching studies. Section 3 explains the holistic matching of strokes, and Section 4 presents the partial matching of strokes. Section 5 reports the design of a pipeline matching experiment and evaluates the results. Section 6 presents the conclusions of this research.

## 2. Related Work

Data matching is a relevant research topic in the geographic information system (GIS) field, with many direct and indirect applications. Among them are data integration, conflation, quality evaluation, and data management [15]. One of the initial matching approaches was developed by Saalfeld [11]. Saalfeld proposed a road network matching method based on buffer growth. The method uses the buffer of the to-be-matched segment to determine a candidate matching set; then, the candidate segments are assessed. Since then, many efforts have been made to develop object-matching methods. In general, the existing predominant matching algorithms address two issues: the type of similarity measure used to determine corresponding objects, and the data matching strategies for decision-making.

To address the first issue (determining the type of similarity measures), Chehreghan [18] studied methods for evaluating the spatial similarity in aspects such as distance, angle, area, and shape. Distance is the most widely used geometric measure for evaluating the matching degree of two spatial targets [19,20]. In the matching process, the distance between a point and a polyline [16] or the Hausdorff distance between two polylines can be used [7,8]. In addition to using distance to evaluate the geometric similarity between two spatial line elements, one or more measures—including the angle, node topological relation [21–23], area, or shape [24]—must be combined to improve the reliability of target matching [9,10,25,26]. Considering the spatial context as a broader concept beyond matching pairs, Samal et al. applied landmarks for similarity assessment. By combining multiple metrics, such as the position and an attribute, the authors proposed the use of landmarks to build a proximity graph, which is a weighted directed graph that is defined to assess the similarity in geographical contexts on the basis of the "proximity" relative to some pre-selected landmarks. The similarity is measured using the total vector offset of the corresponding objects in both datasets [27].

In addition, many scholars have combined geometric characteristics with the spatial relationships in the neighborhood space or the overall pattern [28,29]. Many studies have employed residential areas in the surrounding neighborhood for constraint matching. Wang et al. [30] determined the residential groups adjacent to the road by constructing a skeleton network. Then, the spatial relationship and geometric similarity of the residential groups were calculated and used to determine the matching of the corresponding roads. Fan et al. [31] employed the matching of polygons formed by roads (which can be regarded as residential areas) to indirectly perform road matching. A road has both grades and different levels of semantic features. Matching only the roads at the same level can reduce the search range of the candidate matching set in the matching process. Hu et al. [32] divided road data into three levels, including line segments, road sections, and paths. The buffer and topological relationships were used during road matching at different levels. In their approach to classifying road strokes, Liu et al. [33] first selected high-grade backbone roads as matching objects. Then, they performed

stepwise iterations so that the matching information was transferred in the network model to obtain the matching results.

The second issue concerns the strategy of matching. Walter and Fritsch [10] proposed a mutual information function for segment matching using the buffer algorithm, in which the mutual information is not a cost function but a merit function. This means that matching is possible without the use of data that depend on tuning factors. The buffer is used to search for potential matching objects, and then the mutual information function is used to calculate the shape, position, topological relationship, and other characteristics of the line objects. Finally, the maximum probability is taken as the final matching result. Mantel [34] improved the search speed of the candidate matching set by adding semantic information to the basis of the buffer growth method, and Zhang et al. [35] proposed an algorithm that can dynamically adjust the buffer radius for matching street data. Additionally, probability relaxation has been used to match road networks [36–40]. First, the initial matching matrix is established, and then geometric measures, such as the length, angle, and shape, are calculated using the structural relations of the target in the adjacent areas of the road. The matrix is iteratively updated to achieve the global optimum. Logistic regression is another mathematical model that has been applied for road matching. Tong et al. [41] first described the difference in roads through geometric feature information of the roads. Then, initial matching results were obtained using the "opt method" algorithm. The missed matching and incorrect matching are identified by the optimization and iterative logistic regression matching (*OILRM*) algorithm. In addition, the regression model can be directly constructed using the geometric feature information to predict the matching probability [42]. Alternatively, it can be constructed by training the feature samples through a support vector machine (*SVM*). In the mesh vector data composed of linear elements, the topological relationship between intersection and arcs has important matching information. In [43] and [44], the authors used the matching between nodes and the matching between lines to find 1:1 and 1:N matches, which can solve the problem of node topological changes. Then, they divided the road matching into segment matching and node matching.

The previous research results of road network matching provide a very good baseline for pipeline matching, but there are few reference studies on pipeline matching. Using semantic similarity, Gong et al. [5] presented the pipeline-point ontology concept, and additionally introduced shape similarity characterized by the shape of arcs between two pipeline points. However, semantic information is often inconsistent or missing, and the accuracy of the matching results depends on whether the semantic information is comprehensive. In addition, the problems (growth, shortening, topological changes, etc.) in partial matching between line individuals cannot be solved with the "node–segment" spatial data organization method, so a partial matching algorithm was developed in our study. The matching of underground pipeline networks requires consideration of changes in pipeline morphology and topology, as well as frequent N:M matching types. There are generally four cases of object-matching relationships: one-to-one (1:1), one-to-none (1:0), one-to-many (1:N), and many-to-many (N:M) [16]. Figure 2 shows the possible instances of matching.

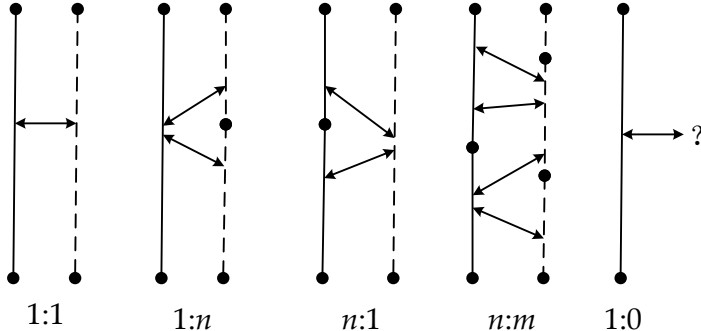

**Figure 2.** Cardinality of the matching pairs.

In cases of data updates (growth, shortening, emergence, disappearance, etc.), there may be changes in same-name objects between the integrated and professional underground pipeline networks. For example, one pipeline in the integrated pipeline map may split into two pipelines in the professional pipeline map, and part of the pipeline may disappear on account of the updates. In this case, a stroke formed by connections based on angle differences may lead to stroke mismatching. This type of stroke mismatching is due to changes in same-name objects, such as a significant difference in length. Thus, we attempt to improve the matching methods of underground pipeline networks in order to deal with these changes in pipelines. The approach developed here encompasses several steps. In the first phase, the data are prepared to allow for an optimized matching process. Then, the matching is performed in several iterations to achieve the final result, which is stored as explicit relations between corresponding features. The resulting set also includes features for which no matching candidates could be found.

## 3. Holistic Stroke Matching

In the matching process, only the geometric properties of a stroke are of concern, and other strokes with direct or indirect topological connections within the stroke's neighborhood or even the entire underground pipeline network are ignored. In this case, the reliability of the matching will be lacking, and other strokes with topological relations will often have inconsistent matching results. That is, the matching between strokes cannot take advantage of neighboring strokes or even the whole underground pipeline network as constraint conditions. When a person reads a map and searches for a pipeline, he or she not only identifies the geometry of the target pipeline but also refers to whether other geographical elements around the pipeline are consistent with the real-world situation. Similarly, in the matching of underground pipelines, in addition to calculating the geometric similarity between strokes, matching references are required. The matching of strokes can only be judged as a whole by checking the matching of surrounding underground pipelines, instead of relying on the local judgment of independent geometric similarity between strokes. Therefore, in the process of matching underground pipelines, the geometric similarity between stroke pipelines and the structural similarities of the networks (neighborhood space) in which the pipelines are located should be calculated to obtain more reliable matching results. This method is called holistic stroke matching.

### 3.1. Geometric Similarity

In holistic stroke matching, a stroke is used as the basic matching unit during pipeline network matching between integrated pipelines and professional pipelines. By calculating the geometric similarity between a stroke and a candidate matching object, a more accurate matching object can be obtained. Geometric measures are used to quantify the magnitude of certain morphological characteristics of the stroke in the geographical space. The main geometric measures are length, angle, shape, area, and distance, among others. However, it is not necessary to include all of the geometric measures for geometric similarity. To obtain an efficient measurement method of geometric similarity while also ensuring effectiveness—that is, the quantified geometrical similarity conforms to human spatial cognition rules—three geometric measures are proposed in this work: length, angle, and spatial distance.

Since a stroke is composed of a number of discrete segments, there is a high degree of differentiation in the lengths of different strokes. Many studies have used length as a matching geometric measure. Furthermore, the angle between the two strokes to be matched is defined as the angle between the two lines connecting the first and last nodes of the two strokes. Generally, the length of a stroke is not short. Thus, for the cases where the lengths meet the required criteria, it is rare that the angle formed by the lines connecting the first and last nodes also meets its required criteria. Hence, spatial distance is an indispensable geometric measure in addition to length and angle; the higher the proximity of the two strokes, the greater their association.

To characterize the spatial distance, typically the Hausdorff distance (*HD*) is employed, which is a mathematical construct that measures the "closeness" of two sets of points that are subsets of a metric

space. Given two point sets, $p_A = \{p_{a1}, p_{a2}, \dots, p_{am}\}$ with size $m$ and $p_B = \{p_{b1}, p_{b2}, \dots, p_{bn}\}$ with size $n$, the classical *HD* between the two point sets is given by:

$$HD(p_A, p_B) = \max\{h(p_A, p_B), h(p_B, p_A)\}, \tag{1}$$

where $h(p_A, p_B)$ is the maximum of the minimum Euclidean distances from each point in point set $p_A$ to point set $p_B$, and vice versa for $h(p_B, p_A)$.

　　In this paper, the spatial distance between strokes is calculated using a modified Hausdorff distance (*MHD*) algorithm, which is insensitive to noise [45]. The *MHD* algorithm is calculated as:

$$\begin{cases} MHD(p_A, p_B) = \max(h(p_A, p_B), h(p_B, p_A)) \\ h(p_A, p_B) = \frac{1}{m}\sum_{p_{ai}\in p_A} \min_{p_{bj}\in p_B} \|p_{ai} - p_{bj}\| \\ h(p_B, p_A) = \frac{1}{n}\sum_{p_{bj}\in p_B} \min_{p_{ai}\in p_A} \|p_{bj} - p_{ai}\| \end{cases}, \tag{2}$$

where $h(p_A, p_B)$ is the average value of the minimum Euclidean distances from each point in point set $p_A$ to point set $p_B$, and vice versa for $h(p_B, p_A)$, $\|p_{ai} - p_{bj}\|$ is the Euclidean distance between $p_{ai}$ and $p_{bj}$. *MHD*(∗) is the *MHD* algorithm.

　　The range of geometric similarity between strokes is [0, 1]. The measurement process is shown in:

$$\begin{cases} SI(s_1, s_2) = \omega_1 * \frac{HT - H(s_1, s_2)}{HT} + \omega_2 * LR(s_1, s_2) + \omega_3 * \frac{AT - A(s_1, s_2)}{AT} \\ if\ H(s_1, s_2) > HT\ then\ H(s_1, s_2) = HT \\ if\ A(s_1, s_2) > AT\ then\ A(s_1, s_2) = AT \end{cases}, \tag{3}$$

where the three experience weight parameters, $\omega_1$, $\omega_2$, and $\omega_3$ are positive numbers, and $\omega_1 + \omega_2 + \omega_3 = 1$. Here, $s_1$ and $s_2$ represent two strokes, *SI*(∗) is the geometric similarity between $s_1$ and $s_2$, and $H$(∗) is the *MHD* algorithm. In addition, *HT*(∗)denotes the Hausdorff distance threshold; *LR*(∗) represents the length ratio between two strokes; $A$(∗) is the angle between the two strokes (the angle formed by the two lines connecting the first and last nodes); and *AT*(∗) is the angle threshold.

　　Discrete segments are handled by stroke connection. The resulting strokes are typically long and thus, length is a key geometric measure. In the experiments below, the value of $\omega_1$ was selected to be 0.5. The Hausdorff distance is not only able to measure the spatial distance between strokes but also indirectly measures the geometric similarity, which is a relatively comprehensive measure. Thus, the value of $\omega_2$ was selected to be 0.35. Finally, the angle is a geometric measure that indicates direction, and the value of $\omega_3$ was selected to be 0.15. In the measures of geometric similarity, if the calculated Hausdorff distance between the two strokes is higher than a threshold, then we interpret this to mean that there is no partial matching relationship in spatial distance similarity. Similarly, if the calculated results of the angles of the two strokes are higher than the threshold, then we infer that there is no partial matching relationship in angle similarity.

　　After calculating the geometric similarity, candidate matching pairs are identified as the pipeline pairs with a geometric similarity that exceeds a given threshold, and then their structural similarity is calculated to further assure that the matching is correct.

### 3.2. Structural Similarity

　　Geometric characteristics between strokes often lack consideration of the overall underground pipeline networks. Moreover, many studies have heavily depended on the geometric similarity between line individuals [6–11]. In fact, each stroke within a pipeline network has a certain topological relationship with the surrounding strokes. The stroke's spatial scene is a local network formed by strokes in the neighborhood that have a direct relationship with the given stroke. Figure 3 shows that stroke pipeline *a* (red) directly connects with several strokes; *b*, *c*, *d*, *e*; and thus they have a direct topological relation. They constitute the spatial scene of *a*. A spatial scene comprises a collection of

spatial objects and their particular arrangement. People construe such spatial scenes in terms of objects and relations [46].

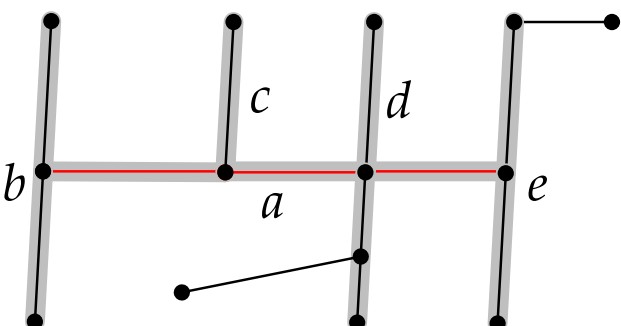

**Figure 3.** Spatial scene of the stroke *a* (red).

The stroke type in a spatial scene can be further categorized. Discrete segments are handled by stroke connections, and the resulting strokes are typically long. Strokes with adjacent topologies can be divided into two categories: (1) Type A, which refer to strokes connected to the start and end nodes, and (2) Type B, which refer to strokes connected to the inner region. For example, Figure 4a is a schematic diagram of stroke pipeline *S* and its spatial scene; Figure 4b shows the Type A strokes in the spatial scene; and Figure 4c shows the Type B strokes in the spatial scene. These two types of pipelines together constitute the spatial scene of *S*.

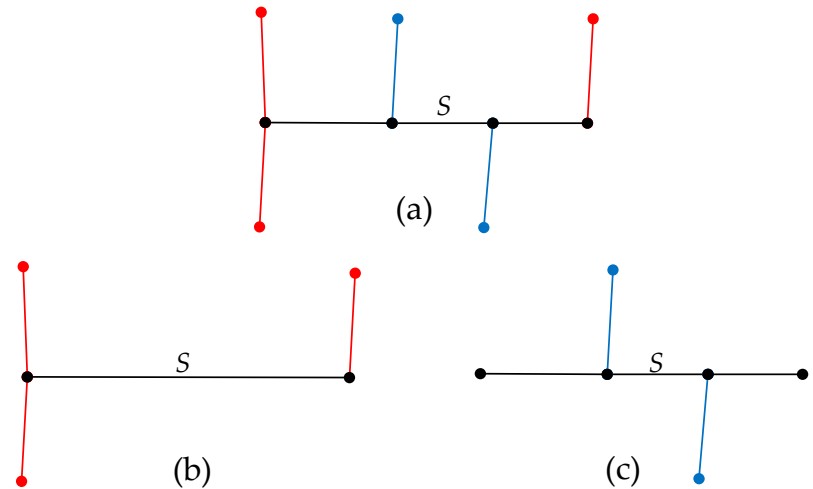

**Figure 4.** Stroke classification in the stroke spatial scene. (**a**) Shows the spatial scene of S; (**b**) shows the Type A strokes; (**c**) shows the Type B strokes.

Let us suppose that a certain stroke is denoted as *S* in the integrated pipeline map, which has two types of neighboring stroke datasets with a direct topological relationship, namely Type A and Type B. Similarly, a candidate stroke matching *S* in the professional pipeline map is denoted *S'* and Type A' and Type B' strokes in the spatial scene of *S'* are characterized. The strokes in the spatial scenes of from *S* and *S'* can be classified as matching or non-matching pairs, respectively, by calculating their geometric similarity using Equation (3). The structural similarity between the spatial scene of *S* and *S'* is then calculated by:

$$SIM(S, S') = f * \frac{\sum\limits_{i=1}^{n} len(i) * SI(i) + \sum\limits_{i=1}^{m} len(i) * SI(i)}{\sum\limits_{i=1}^{n} len(i) + \sum\limits_{i=1}^{m} len(i)}, \tag{4}$$

where $len(i)$ is the sum of the lengths of the two matching strokes in the $i$-th matching pair; and $SI(i)$ is the geometric similarity between the two matching strokes in the $i$-th matching pair. Furthermore, $n$ is the total number of matching pairs of Type A and A′, and $m$ is the total number of matching pairs Type B and B′.

The compensation function $f$ is added to Equation (4), and it is defined as:

$$f = \frac{(n+m)*2}{T},$$ (5)

where $T$ is the total number of strokes in the two spatial scenes of $S$ and $S'$, and $n + m$ is the number of successfully matched stroke pairs between the two spatial scenes. The effect of the compensation function is that the higher the number of strokes matched in the two spatial scenes, the higher the structural similarity of the two spatial scenes.

In Figure 5, strokes $A$, $B$, and $C$ are two strokes from one dataset, while $A_1$, $B_1$, and $C_1$ are from the other dataset. Without calculating the structural similarity, we may mistakenly identify $B$ and $A_1$ as a matching pair. This type of error can be prevented by calculating the structural similarity.

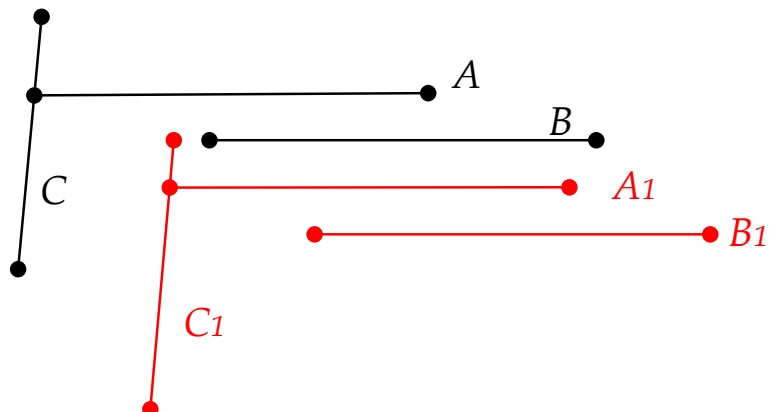

**Figure 5.** The importance of structural similarity. Strokes $A$, $B$, and $C$ are two strokes from one dataset (black line), while $A_1$, $B_1$, and $C_1$ are from the other dataset (red line).

Both the geometric and structural similarity are indispensable for achieving an accurate and robust result. Based on the fact that the calculating methods for structural similarity are complex and time-consuming, it is impossible to calculate the structural similarity of all the pipelines in the dataset. Therefore, we calculate the geometric similarity first, and structural similarity is only calculated for the objects whose geometric similarity is higher than the threshold. As the numbers of links in strokes are different, a stroke may contain one or several links. In Figure 6a, $AC$ and $A_1C_1$ are the matched strokes, and $AC$ has two links—$AB$ and $AC$—while $A_1C_1$ has only one link, which is a one-to-many matching type. If $A_1C_1$ were composed of three or more links (as shown in Figure 6b), our algorithm could also find a many-to-many matching type. After holistic matching, the one-to-one, one-to-many, and many-to-many matching types (1:1, 1:$n$, $n$:$m$) can all be identified.

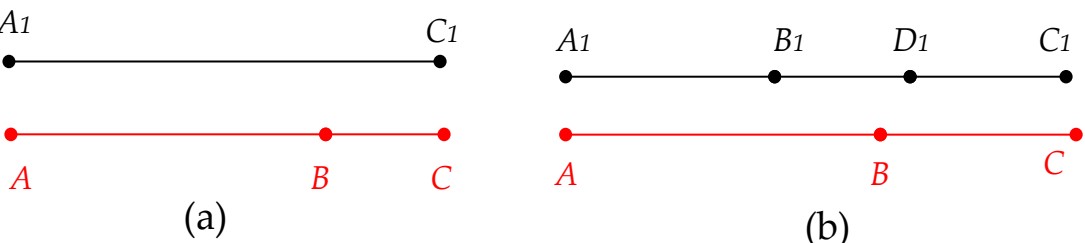

**Figure 6.** (**a**) Shows the one-to-many conditions; (**b**) shows the many-to-many conditions.

## 4. Partial Stroke Matching

*4.1. Stroke Partial Matching Algorithm Based on Segment Decomposition (SPMA-S)*

After holistic stroke matching, a partial pipeline in a stroke could be matched with pipelines in another map. Thus, we need a method to deal with these pipelines. The data processing for such a scenario is shown in Figure 7.

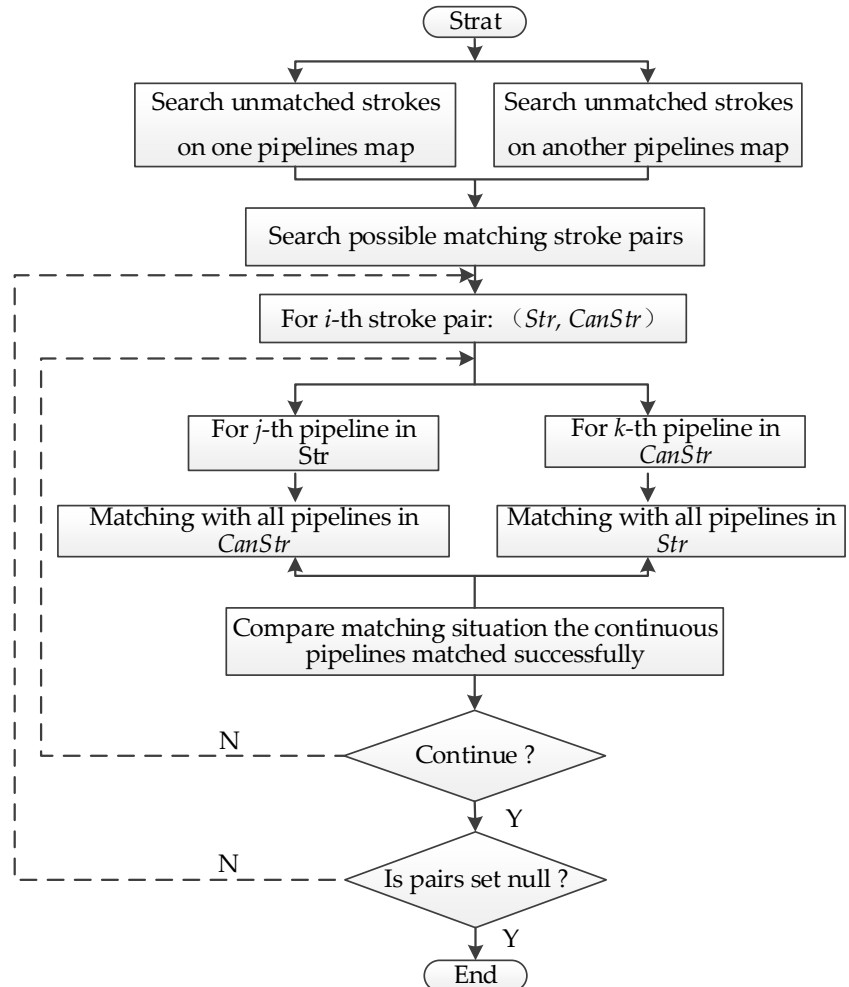

**Figure 7.** Data processing for partial stroke matching.

The map data of one pipeline (such as an integrated pipeline) are denoted as the set *Str* and the map data of the other pipeline (such as a professional pipeline) are denoted as set *CanStr*. *Str* is composed of $N$ pipelines, i.e., $Str = \{Str_1, Str_2, \ldots\ldots Str_N\}$, and *CanStr* consists of *NC* pipelines, i.e., $CanStr = \{CanStr_1, CanStr_2, \ldots\ldots CanStr_{NC}\}$. In the calculation process, we need to specify the Hausdorff distance threshold *HT*. In our experiment, the algorithm proposed in [8] is particularly suitable for calculating the spatial distance between two linear features with large differences in length. After we obtain the spatial distance *HD* between each pipeline of $Str_i$ and $CanStr_j$, we determine whether each segment of $Str_i$ can find the corresponding part in $CanStr_j$, If *HD* is greater than *HT*, then the spatial distance attribute value $HAtt_i$ of $Str_i$ is *FALSE*, which means that there is no partial matching between the pipeline of $Str_i$ and $CanStr_j$. Otherwise, $HAtt_i$ is *TRUE*. After calculation, the Boolean spatial distance attribute values of each pipeline of *Str* form a set, $PStr_i$:

$$PStr_i = \{HAtt_1, HAtt_2, \cdots\cdots, HAtt_N\}. \tag{6}$$

Similarly, the spatial distance *HD* from $CanStr_j$ (where $j$ = 0, 1, ......, NC) to *Str* is calculated, and $HAtt_j$ can be obtained accordingly. This forms the Boolean set $PCanStr_j$:

$$PCanStr_j = \{HAtt_1, HAtt_2, \cdots\cdots, HAtt_{NC}\}. \tag{7}$$

The values of the elements of *PStr* and *PCanStr* are compared and analyzed. If the element values of some continuous pipelines are always *TRUE* from left to right, then a matching body is formed. A so-called matching body means that the matching object of a certain part of the stroke has been found in another pipeline map. These parts of strokes in *Str* or *CanStr* are then connected to form a new stroke. At this point, preliminary results of the partially matched internal sub-strokes in *Str* and *CanStr* are obtained.

Finally, the length ratio and angle between newly-generated strokes are calculated. If both the length ratio and the angle meet the threshold conditions, then the partial matching is considered successful. All the pipelines with successful partial matching are marked as matched. If one of the length ratios or the angle does not meet the threshold conditions, then one pipeline of the longer stroke is removed in accordance with the order (from first to last). The remaining pipelines form a new stroke. Then, the length ratio and the angle with the shorter stroke in the matching pair are calculated, until the threshold conditions are met or all the pipelines are completely removed.

### 4.2. Stroke Partial Matching Algorithm Based on Vertex Decomposition (SPMA-V)

The matching pipelines are connected by red arrows in Figure 8. With the urban underground pipeline transformation, the topology of the pipelines may change for different survey purposes and dates of formation and measurement of the integrated and professional pipelines.

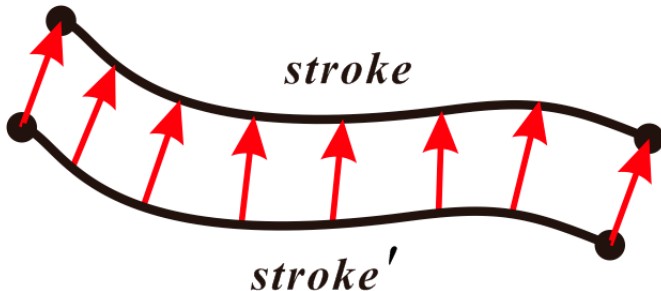

**Figure 8.** Arrow visualization of matching results. Here *stroke* is the reference data, *stroke'* is the target data, and the matching direction is *stroke* to *stroke'*. The arrows that match the result are marked *stroke* with *stroke'*.

In Figure 9a, one pipeline map is $S$ = {$a_1$, $a_2$, $a_3$} and another is $S'$ = {$b_1$, $b_2$, $b_3$, $b_4$}. In the algorithm, to retain the original data as much as possible, we first performed partial stroke matching based on segment decomposition (*SPMA-S*). Thus, we obtained a partial match between $a_2$ and $b_2$, $b_3$, as shown in Figure 9b. A partial matching relationship in the spatial distance and approximate shape exists for $a_3$ and $b_4$, as well as for $a_1$ and $b_1$. However, they cannot be completely considered to be a partial matching relationship according to *SPMA-S*. In order to solve the problem in Figure 9b, we decomposed the strokes by vertices (red dots in Figure 9c). Then, we entered the stroke partial matching algorithm based on vertex decomposition (*SPMA-V*) to search for the relation of partial pipeline matching to a greater extent. The basic unit of the *SPMA-S* algorithm is the pipeline segment. The decomposition algorithm is very good by splitting edges by transferring nodes from one representation to the other [25]. In our algorithm, we considered simplifying the complexity of the algorithm, and we chose the average interpolation method with fixed length. In the partial stroke matching *SPMA-V* algorithm, every unmatched stroke is decomposed by inserting vertices according to a certain vertex density. The basic matching unit is thus refined to the vertex of the stroke. As shown in Figure 9c, the matching result of

the vertex is used to obtain a partial matching result. In reality, the growth, shortening, appearance, and disappearance of pipelines often occur, and the *SPMA-V* is more effective than the *SPMA-S* algorithm when such changes occur. Furthermore, the problems (growth, shortening, topology changes, etc.) in partial matching between individual lines can be solved.

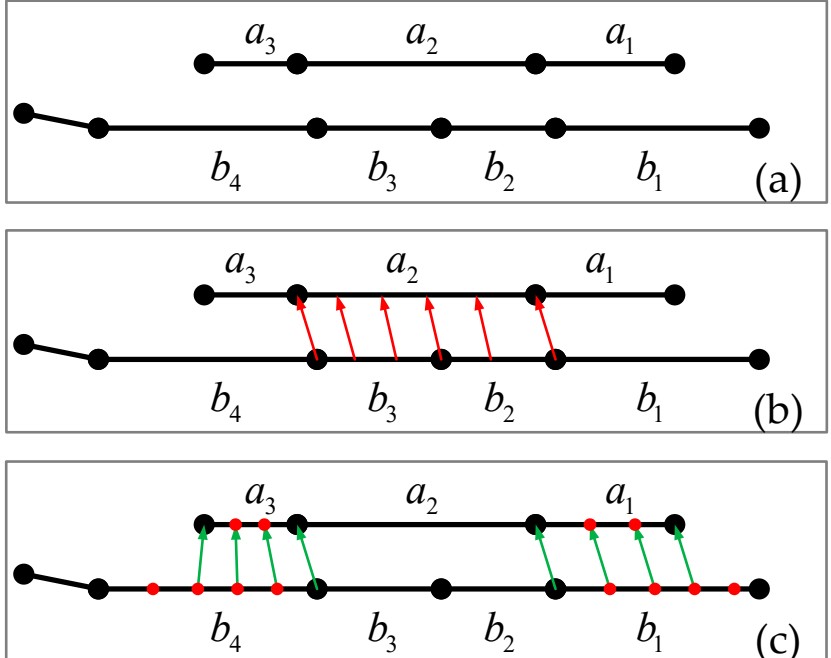

**Figure 9.** Complementary relationship between two partial stroke matching algorithms: (**a**) shows two initially unmatched strokes; (**b**) shows the result of the *SPMA-S* algorithm (red arrows represent the matching relationship); (**c**) shows the result of the *SPMA-V* algorithm (green arrows represent the matching relationship, and red dots are the inserted vertices according to a certain density).

To formalize the *SPMA-V*, approach, assume a certain stroke pipeline is composed of $k$ segments and denoted as $S = \{s_1, s_2, ......, s_k\}$. Furthermore, assure there are $n$ vertices in the stroke, such that the set of ordered vertices of the stroke is $SN = \{p_1, p_2, ......, p_n\}$. Similarly, the candidate matching pipeline stroke on another pipeline map is denoted as $S' = \{S'_1, S'_2, ......, S'_{k'}\}$. There are $m$ vertices in this stroke, and the ordered vertex set is $SN' = \{p'_1, p'_2, ......, p'_m\}$. Each vertex in $S$ and $S'$ has a Boolean spatial distance attribute value (similar to the *SPMA-S* algorithm).

First, the shortest distance from each vertex $p_i$ (where $1 \leq i \leq n$) in $S$ to $S'$ is obtained, and the spatial distance attribute $attr_i$ of the vertex $p_i$ is determined according to the shortest distance via:

$$attr_i = \begin{cases} TRUE & if \min\left(\left\|p_i - p'_j\right\|\right) \leq HT \\ FALSE & if \min\left(\left\|p_i - p'_j\right\|\right) > HT \end{cases} \quad p'_j \in SN', \tag{8}$$

where $\|\ldots\|$ represents the Euclidean distance between two points, and $HT$ is the spatial distance threshold. Thus, the Boolean spatial distance attribute value set for $SN$ is obtained and can be expressed as:

$$SNAttr = \{attr_1, attr_2, \cdots\cdots, attr_n\}. \tag{9}$$

Similarly, the shortest distance is obtained for each node $p'_j$ (where $1 \leq j \leq m$) of $S'$, and the Boolean spatial distance attribute value is obtained by:

$$attr'_j = \begin{cases} TRUE & if \min\left(\left\|p'_j - p_i\right\|\right) \leq HT \\ FALSE & if \min\left(\left\|p'_j - p_i\right\|\right) > HT \end{cases} \quad p_i \in SN. \tag{10}$$

Finally, the Boolean spatial distance attribute value set for $SN'$ is obtained and is:

$$SNAttr' = \{attr'_1, attr'_2, \cdots\cdots, attr'_m\}. \tag{11}$$

The elements of *SNAttr* and *SNAttr'* are combined to form a matching body. If the indices of the elements continuously vary from small to large, and the corresponding spatial distance attribute values are *TRUE*, these continuous ordered vertices form a matching body. Matching bodies with the same index in *SNAttr* and *SNAttr'* are combined to form a matching pair. The corresponding ordered vertexes in the matching pair are then combined into the stroke, and partial matching results of the stroke are obtained.

## 5. Experiment

### 5.1. Experimental Data and Matching Process

As an experimental test of our approach, we collected gas pipeline data from a location in the city of Wuhan, in the Hubei province of China. The integrated pipeline data and professional pipeline data are respectively from the municipal surveying and mapping department and the gas company. The integrated pipeline (gas) and professional pipeline (gas) data are summarized in Table 1. In our experiments, Visual Studio 2010 (C#) was chosen to design and implement the algorithm. The matching algorithm was used to find the relationship between entities with the same name between the integrated pipeline (gas) data and the professional pipeline (gas) data. If data are captured repeatedly systematic errors and local differences will arise through digitization. Systematic errors were introduced, for example, by inaccurate transformation between digitizing tablet and model coordinate system. Local differences are the result of capturing data in different data models, as well as incorrect coordinates, topology errors, or missing elements. A transformation has to be computed to minimize the global error. The parameters of the transformation were calculated by measuring control points interactively on the screen. A further automation is possible by finding corresponding points in two data sets automatically.

**Table 1.** Summary of the experimental data from the integrated and professional pipelines.

| Study Data | Data Sources | Data Format | Number of Pipelines | Total Length (m) |
|---|---|---|---|---|
| Integrated pipeline (gas) | Institute of surveying and mapping | MDB | 1138 | 11,255.573 |
| Professional pipeline (gas) | The gas company | MDB | 1274 | 11,966.123 |

Figure 10a shows the integrated pipeline (gas) map, Figure 10b shows the professional pipeline (gas) map, and Figure 11 shows the superposition of the two data sets. As can be seen from the superposition map, although the data of the two pipelines basically overlap, there are still obvious differences in the details. The differences are mainly due to the various matching relationships, such as 1:1, *n*:1, and *n*:*m*, that exist between the two types of pipeline data in pipe segment matching. Statistics for some matching types are shown in Table 2. Given this data, it is difficult to perform accurate matching by using only the spatial similarity method, and it is further difficult to distinguish multiple similar pipeline entities using the uniform method. Thus, we added structural similarity in addition to spatial similarity in order to more effectively match dissimilar entities in a similar region. For unmatched pipelines, we executed partial matching methods which include both the stroke partial matching algorithm based on segment decomposition (*SPMA-S*) and stroke partial matching algorithm based on vertex decomposition (*SPMA-V*).

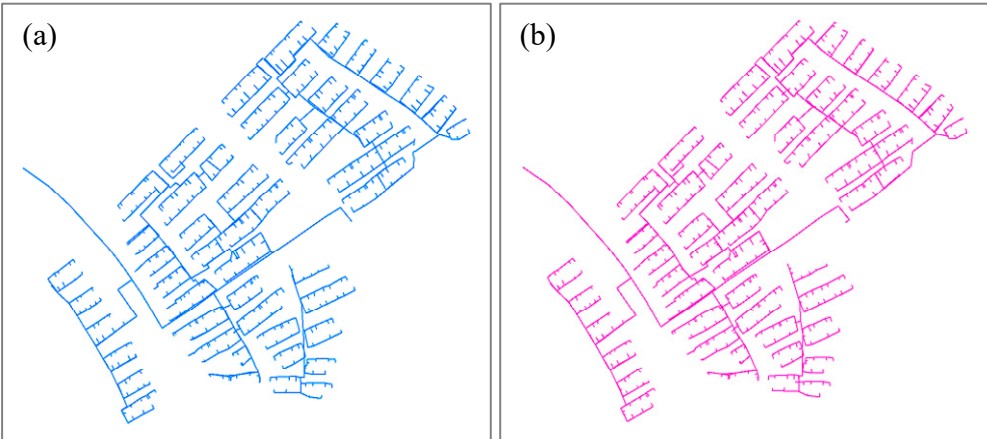

**Figure 10.** (**a**) The integrated pipeline (gas) map with pipelines shown in blue; (**b**) the professional pipeline (gas) map with pipelines shown in pink.

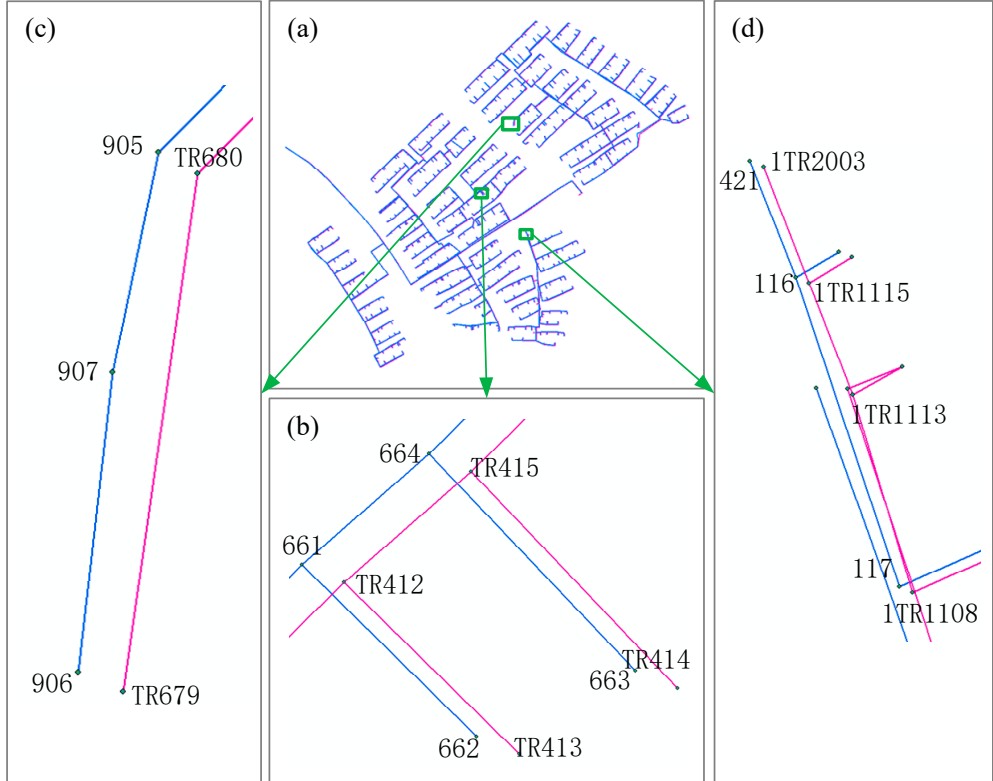

**Figure 11.** The superposition of the integrated and professional pipeline maps. The blue lines represent integrated pipelines, and the pink lines represent professional pipelines. (**a**) Shows the superposition; (**b**) shows the 1:1 Matching Type; (**c**) shows the *n*:1 Matching Type; (**d**) shows the *n*:*m* Matching Type.

**Table 2.** Example of matching types in the integrated and professional pipelines.

| Integrated Pipeline Segment | Number of Integrated Pipeline Segments | Professional Pipeline Segment | Number of Professional Pipeline Segments | Matching Type |
|---|---|---|---|---|
| 661–662 | 1 | TR412–TR413 | 1 | 1:1 |
| 906–905 | 2 | TR679–TR680 | 1 | *n*:1 |
| 421–117 | 2 | 1TR2003–1TR1108 | 3 | *n*:*m* |

In Table 2 and Figure 11, points 661, 662, 906, 905, 421, and 117 are integrated pipe points and points TR412, TR413, TR679, TR680, 1TR2003, and 1TR1108 are professional pipe points. In both types of pipeline data, pipe segments are represented by start-stop pipe points, and pipelines are composed of multiple pipe segments. The results show the different matching types (1:1, *n*:1, *n*:*m*) of the two pipelines with the same named objects.

The data need to be standardized before algorithm execution. The matching data standards are as follows: (a) the storage format of the two experimental datasets is consistent, and matching experiments can be carried out on a unified platform; (b) the two experimental datasets are matched in a unified coordinate system; (c) both types of experimental data are from pipeline entities within the same range; (d) in the same experimental data, the distance between the starting and stopping points of a pipe segment and the adjacent pipe points are within a certain tolerance.

In addition, a number of matched samples were selected to calculate the distance threshold and angle threshold by spatial statistical analysis. The selection of the sample points needs to ensure that each connected region selects at least one sample point. The average value is determined as the threshold value and is estimated by using the matched samples. The distance threshold is determined to be 4 m, and the angle threshold is determined to be 20° based on the sample data shown in Figure 12.

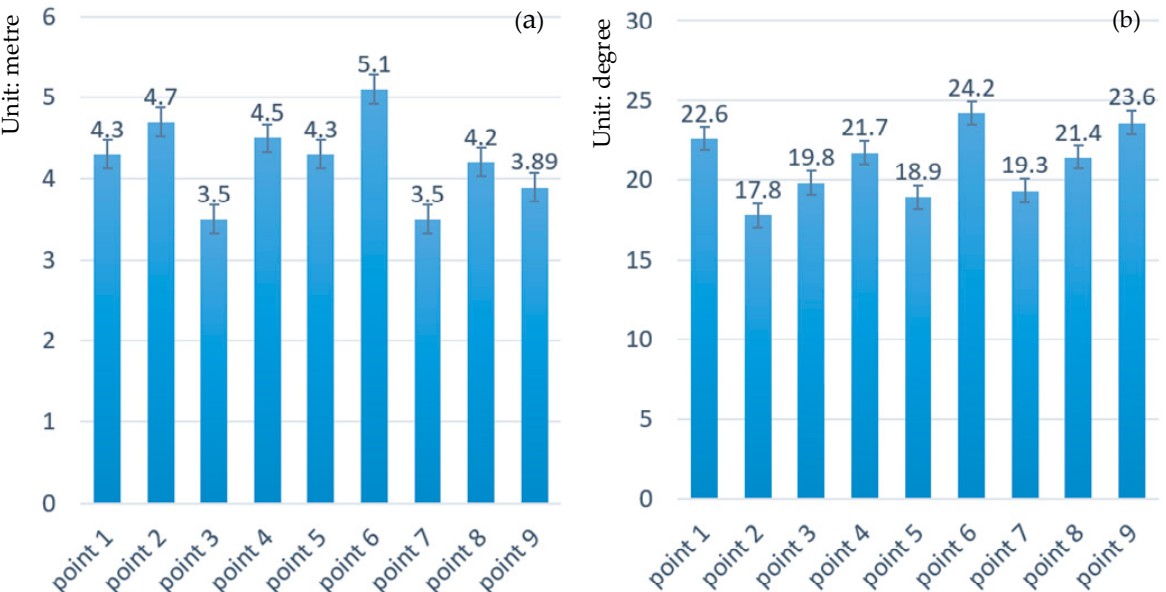

**Figure 12.** (**a**) The Hausdorff distance in meters and (**b**) the angle between the strokes in degrees for a sampling point.

Our matching process employs the following approach and consists of the three steps detailed below. The matching steps of underground pipeline network are shown in Figure 13.

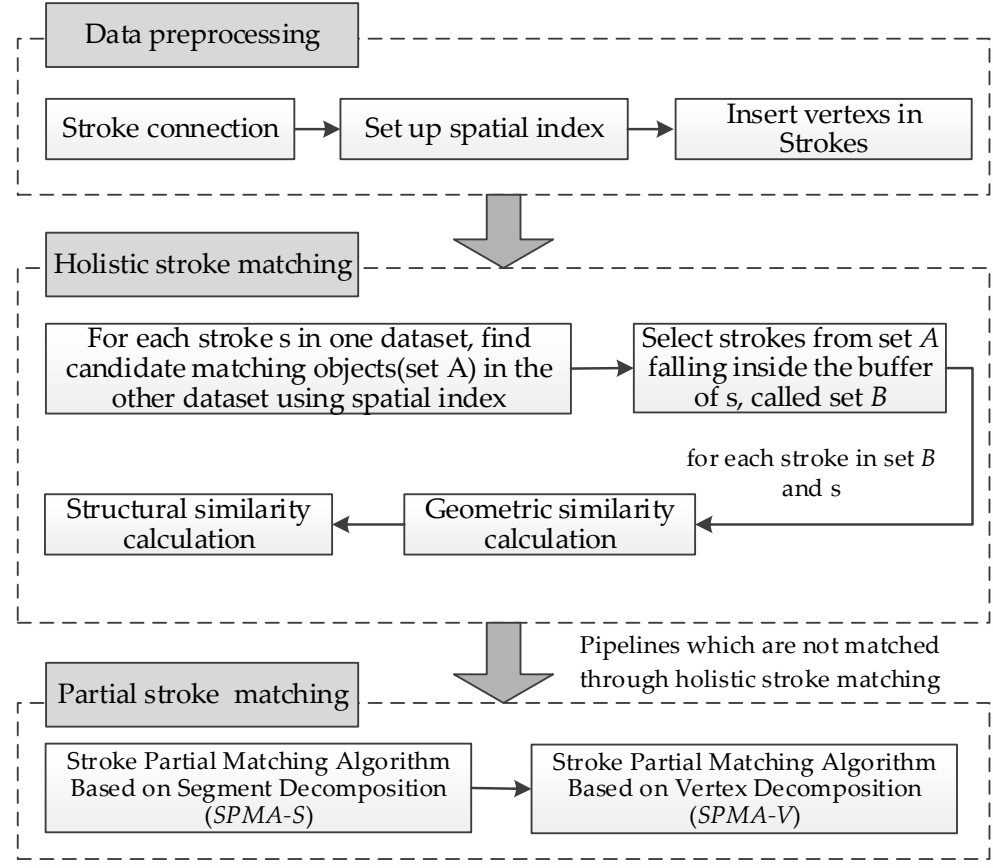

**Figure 13.** The steps of matching underground pipeline networks.

Step 1—Data preprocessing: First, the pipeline segments are connected into strokes using the principle of included angle coherence. Since the vertices of the pipeline are too sparse for the matching process, the calculated spatial distance will be too large to reflect the real differences between pipelines, so the vertices of the pipelines should be encrypted. The average interpolation method with fixed length of stroke is shown in Figure 14. For the purpose of enhancing the efficiency of selecting candidate matching pipelines, spatial grid indexes of the datasets are built, the diagram of spatial grid indexes is shown in Figure 15.

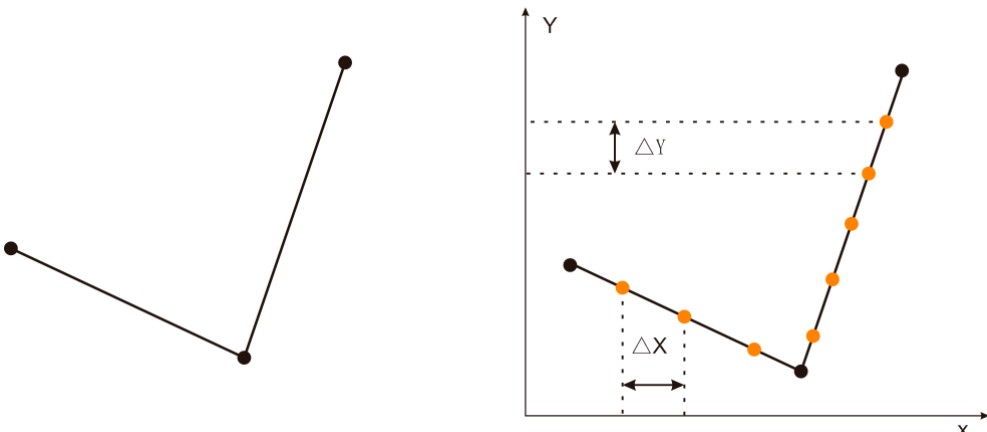

**Figure 14.** Average interpolation method with fixed length of stroke. (**a**) Shows pipeline and (**b**) shows the results after average interpolation method with fixed length.

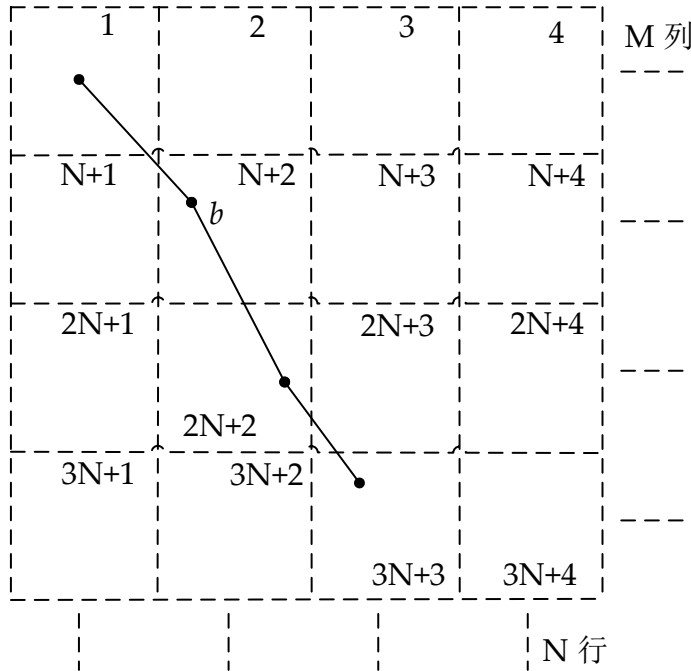

**Figure 15.** The diagram of spatial grid indexes (the grid index numbers covered by stroke *b* are 1, N + 1, N + 2, 2N + 2, 2N + 3, and 3N + 3).

Step 2—Holistic matching of the strokes: For each stroke *S* in one dataset, the candidate matching strokes (denoted as set *A*) are first selected from the other dataset using the spatial index. Then, the strokes in set *A* that fall inside the buffer of *S* are further selected as set *B*, the diagram of approximate buffer is shown in Figure 16. For each stroke $S_1$ in set *B*, the geometric similarity to *S* is calculated. The geometric similarity is the weighted combination of length similarity, angle similarity, and distance similarity, as shown in Equation (3). If the geometric similarity is higher than a given threshold, then we calculate the structural similarity between $S_1$ and *S* to check if their spatial scenes are matched, as shown in Equation (4). Strokes with geometric similarity lower than the threshold are deemed unmatched strokes, and their structural similarity does not need to be calculated. Only the objects whose geometric and structural similarities are both higher than a given threshold are defined as matching pipelines. Others enter the process for the partial matching of strokes.

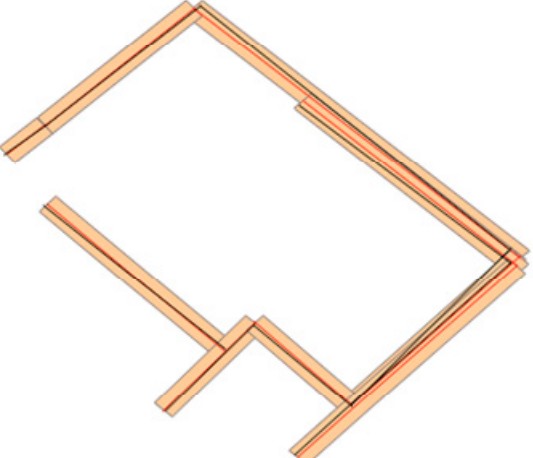

**Figure 16.** The diagram of approximate buffer.

Step 3—Partial matching of strokes. After holistic matching, the overall matching results of integrated and professional pipeline strokes can be obtained, and the unmatched strokes can also be obtained. The partial stroke matching algorithm based on segment decomposition (*SPMA-S*) is designed to maximize the retention of the two sets of original underground pipeline data. After matching with this algorithm, all the pipeline objects that establish a matching relationship are composed of pipeline segments in the original data. In the process of pipeline segment separation and reassembly, if a suitable partial match of an object cannot be found, then the partial stroke matching algorithm based on vertex decomposition (*SPMA-V*) is applied to find other partially matched pipeline segments.

The matching results are shown in Figure 17. Figure 17a shows the matching results for the entire map. Figure 17b shows the results for a typical underground pipeline network, where parallel double pipelines were correctly matched to the corresponding pipelines. Another example is shown in Figure 17c. Here there are two strokes, stroke1<*a*, *b*> and stroke2<*c*, *d*>, in the integrated pipeline map (see Figure 17$c_1$ for details), and three strokes, stroke1<*a'*>, stroke2<*b'*>, and stroke3<*c'*, *d'*>, in the professional pipeline map (see Figure 17$c_2$ for details). In the holistic matching of strokes, stroke2<*c*, *d*> and stroke3'<*c'*, *d'*> are matched to each other. In the partial matching of strokes based on segment decomposition, segments *a* and *a'* are matched, only segments *b* and *b'* do not match because of the large difference in their geometric morphology. Finally, in the partial matching based on vertex decomposition (*SPMA-V*), a matching relationship was determined between the entire pipeline of *b* and part of *b'* (see Figure 17$c_3$ for details). The results indicate that the algorithm described in this paper is accurate and robust, and it identifies numerous matching relationships between different datasets.

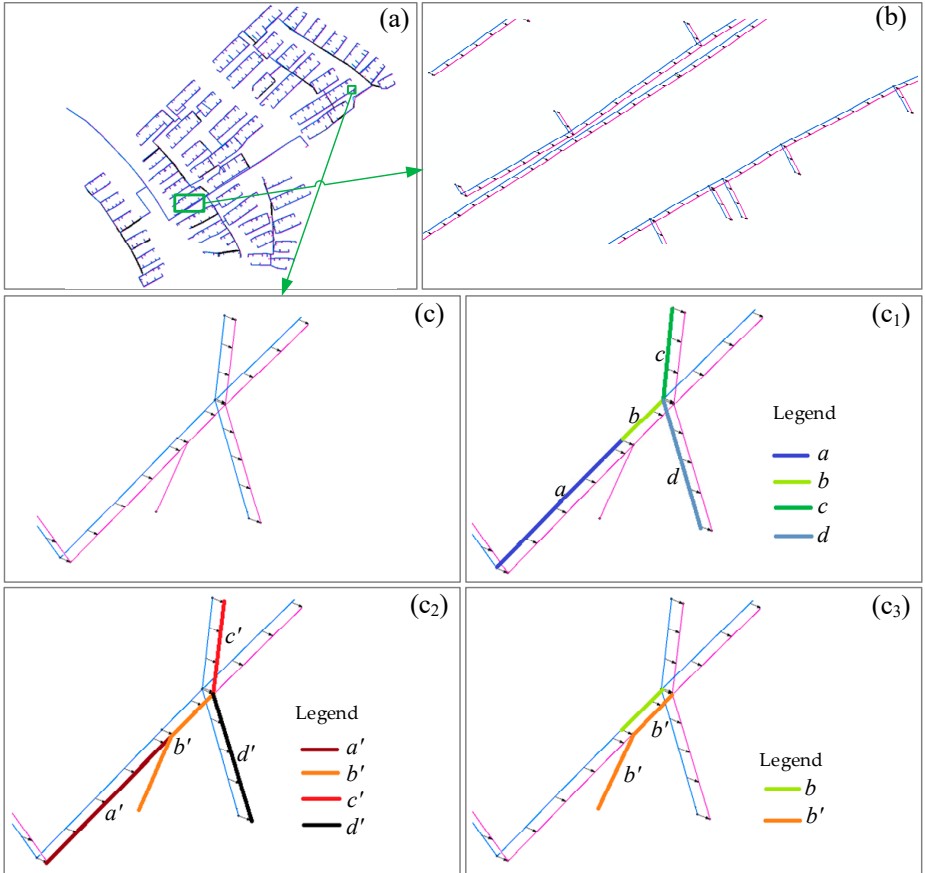

**Figure 17.** Matching results of the integrated and professional pipelines. The lines with arrowheads represent matching relationship, the blue lines are integrated pipelines, and the pink lines are professional pipelines; (**a**) Shows all the matching results; (**b**) shows the matching result of parallel double line; (**c**), (**c₁**), (**c₂**), and (**c₃**) explain the matching process.

### 5.2. Comparative Experiments and Evaluation

We performed underground pipeline matching experiments to compare the performance of our approach with other approaches, including the matching algorithm designed in [36]. The algorithm for the comparison experiment is the matching method for the mesh data of "node–segment", thus, we call it the node–segment algorithm, and we call the algorithm proposed in this paper the stroke algorithm. The results were assessed by the evaluation formula from [41], which is:

$$\begin{cases} P = \frac{f(C)}{f(C)+f(W)} * 100\% \\ R = \frac{f(C)}{f(C)+f(U)} * 100\% \end{cases}, \tag{12}$$

where $P$ is the matching accuracy; $R$ is the matching recall rate; $f(C)$ denotes the number of correct matches; $f(W)$ denotes the number of incorrect matches; and $f(U)$ denotes the number of missed matches.

To evaluate the matching performance, we manually checked the matching results of the experiments by evaluating the matched pairs one by one. Since the data matching approach of the "stroke algorithm" is applied step by step, the statistics of the matching performance in each step are shown in Table 3. Additionally, the statistics of the experiments for the two methods are shown in Table 4.

**Table 3.** Statistics of the matching performance at each step of the "stroke algorithm".

| Algorithm Step | $f(C)$ | $f(W)$ | $f(U)$ | $P(\%)$ | $R(\%)$ | Time Consumed (s) |
|---|---|---|---|---|---|---|
| Holistic Stroke Matching | 578 | 0 | 379 | 100.0 | 60.4 | 2.94 |
| SPMA-S Matching | 192 | 47 | 3 | 80.3 | 98.5 | 1.43 |
| SPMA-V Matching | 187 | 25 | 8 | 88.2 | 95.9 | 1.16 |

**Table 4.** Comparison of the experimental results using two methods.

| Algorithm Type | $f(C)$ | $f(W)$ | $f(U)$ | $P(\%)$ | $R(\%)$ | Time Consumed (s) |
|---|---|---|---|---|---|---|
| Stroke algorithm | 957 | 72 | 11 | 93.0 | 98.8 | 6.53 |
| Node–segment algorithm | 918 | 97 | 25 | 90.4 | 97.3 | 7.95 |

Table 4 shows that the matching accuracy $P$ and recall rate $R$ of the stroke algorithm are better than those of the node–segment algorithm. The main reason for this improvement is that our algorithm organizes data with the stroke as a matching unit. The node–segment spatial data organization method has some obvious shortcomings: (1) it is difficult to analyze and operate the features of linear elements as a whole, and (2) it is difficult to express the changes and distribution of geographical phenomena along linear elements. The first deficiency mainly manifests in the lack of analysis and measurement of the geometric characteristics of the entire pipeline in the matching process. In the underground pipeline network, the matching subject should be the entire pipeline rather than matches formed after analyzing the geometric characteristics of individual segments. The second deficiency is the difficulty measuring the geometric characteristics of pipeline spatial targets. The matching of heterogeneous pipeline data belongs to the category of linear object matching, and there may be matching types that are 1:$n$, $n$:1, or $n$:$m$. If the segment is used as a matching unit, these cases will be difficult to match, where our algorithm addresses the above problems.

Figure 18a is the visualization of the node–segment algorithm result on an example pipeline network in which pipelines $A_1$ ($a_1$) and $B_2$ ($b_3$) are matched. Figure 18b is the visualization of the

stroke algorithm result on the same network in which pipelines $A_1$ ($a_1$) and $B_1$ ($b_1$, $b_2$) are matched, and pipelines $A_2$ ($a_2$) and $B_2$ ($b_3$, $b_4$) are matched. By human judgment, the node–segment algorithm result is an incorrect match, and the stroke algorithm result is a correct match, indicating that the stroke algorithm is more adaptable to topological changes.

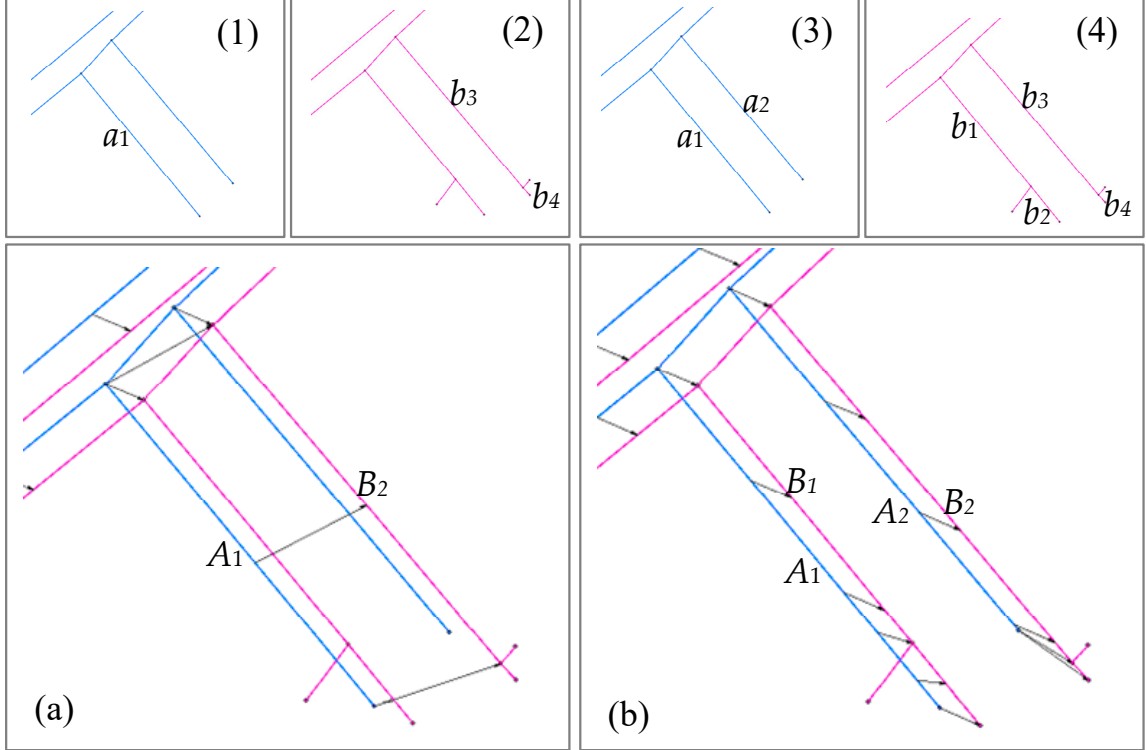

**Figure 18.** A case where the node–segment algorithm leads to an incorrect match, but the stroke algorithm leads to a correct match. The lines with arrowheads represent a matching relationship, (**1**) and (**3**) (blue lines) are integrated pipelines, and (**2**) and (**4**) (pink lines) are professional pipelines. (**a**) Shows the node–segment algorithm result; (**b**) shows the stroke algorithm result.

Figure 19a is the visualization of the node–segment algorithm result for a different example network, in which pipelines $A_1$ ($a_1$) and $B_2$ ($b_3$, $b_4$) are matched. Figure 19b is the visualization of the stroke algorithm result in which pipelines $A_1$ ($a_1$) and $B_1$ ($b_1$) are matched, and pipelines $A_2$ ($a_2$) and $B_2$ ($b_2$) are a partial match. By human judgment, the node–segment algorithm missed a match, and the stroke algorithm missed a match as well, even though there are some partial matches.

Figure 20a,b is the visualization of the stroke algorithm result. Figures 19b and 20a are the same diagram: pipelines $A_1$ ($a_1$) and $B_1$ ($b_1$) are matched, and pipelines $A_2$ ($a_2$) and $B_2$ ($b_2$) are partially matched. Pipelines $A_2$ ($a_2$) and $B_2$ ($b_2$) are partially matched by increasing the distance threshold. This indicates that the stroke algorithm can solve some matching problems that cannot be resolved by changing the threshold value.

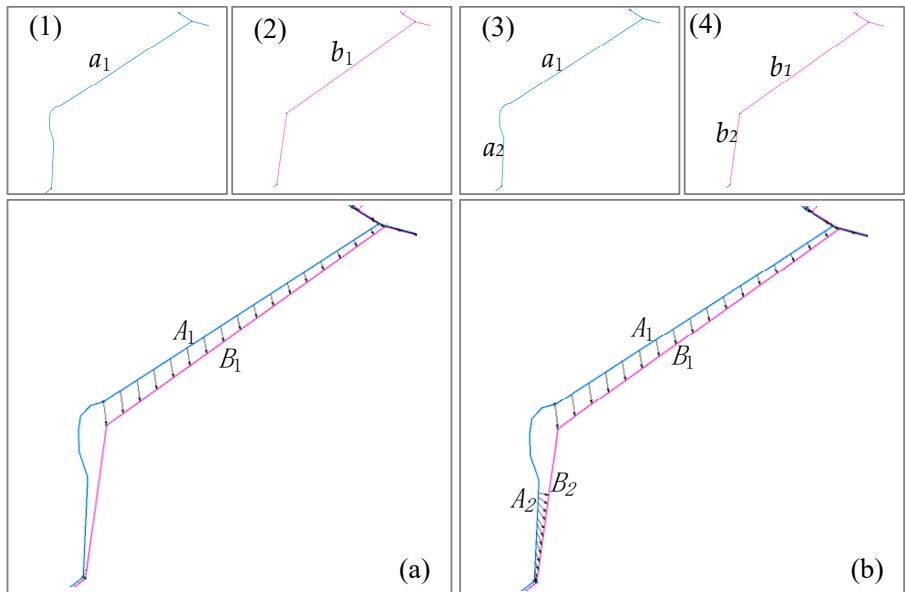

**Figure 19.** The node–segment algorithm missed a match, while the stroke algorithm results in some partial matches. The lines with arrowheads represent a matching relationship, (**1**) and (**3**) (blue lines) are integrated pipelines, and (**2**) and (**4**) (pink lines) are professional pipelines. (**a**) Shows the node–segment algorithm result. (**b**) Shows the stroke algorithm result.

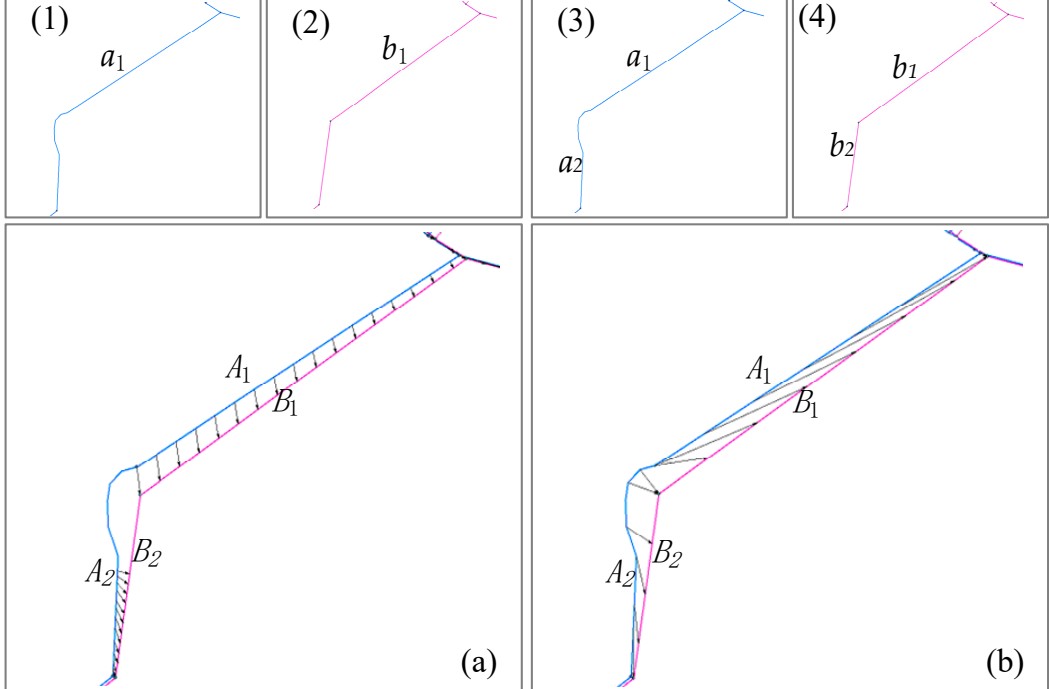

**Figure 20.** The missed match can be optimized to produce a correct match with the stroke algorithm. The lines with arrowheads represent a matching relationship, (**1**) and (**3**) (blue lines) are integrated pipelines, and (**2**) and (**4**) (pink lines) are professional pipelines. (**a**) Shows the stroke algorithm result where the distance threshold is 4 m; (**b**) shows the stroke algorithm result where the distance threshold is 6 m.

## 6. Conclusions

In this paper, we proposed a novel matching algorithm for integrated underground pipeline data and professional underground pipeline data. This approach is a multi-step process, and we

provide statistics on the performance for each step of our algorithm. Our algorithm takes the stroke as the basic unit of matching to retain the characteristics of the smooth and continuous distribution of the source data. In the holistic matching of strokes, we use the length, angle, and spatial distance measures to calculate the geometric similarity, and we also assign different weights and thresholds. In addition, the evaluation method was improved by combining the geometric characteristics with the spatial topological relationships in the neighborhood space, in order to calculate the structural similarity of the spatial scene. The holistic matching of strokes is able to not only solve the matching problem in the conventional case but also better handle 1:$n$ and $n$:$m$ matching situations. As a result, our algorithm overcomes partial failures with the individual similarity measure in the local matching method and has a matching accuracy $P$ of 100%. Since there might be unmatched strokes after holistic stroke matching, a method for partially matching strokes is proposed. The partial stroke matching algorithm based on segment decomposition (*SPMA-S*) is designed to maximize the retention of the original two sets of underground pipeline data. After matching with this algorithm, all the pipeline objects that establish a matching relationship are composed of pipeline segments in the original data. Problems (such as growth, shortening, topology changes, etc.) in partial matching between individual lines can be solved by using the partial stroke matching algorithm based on vertex decomposition (*SPMA-V*). The experimental results show that the improved method has a high matching accuracy, recall rate, and running efficiency.

In our approach, the parameters, including the thresholds and weights, are currently set by human experience. Future research will explore a methodology to automatically set the thresholds and weights according to specific data to further ensure the matching performance and quality. Due to the limited research data, we only used gas pipeline data in a certain area as an example to conduct data matching experiments and did not use other industry pipeline data to verify the pipeline matching method.

**Author Contributions:** Shuai Wang and Qingsheng Guo conceived and designed the experiments; Xinglin Xu and Yuwu Xie performed the experiments; Shuai Wang and Qingsheng Guo analyzed the data and wrote the paper.

**Funding:** This work was supported by the National Natural Science Foundation of China, grant number 41871378, 41471384.

**Conflicts of Interest:** The authors declare no conflict of interest.

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
