# Peer review of "A Study on a Matching Algorithm for Urban Underground Pipelines"

_ijgi, doi:10.3390/ijgi8080352_

Round 1
Reviewer 1 Report
The paper presents a data matching approach to match underground pipelines. The approach is composed on three steps: holistic data matching (based on geometric and structure similarities), partial data matching (based on the decomposition of both segment and vertex). Both holistic and partial data matching are stroke based).
General remarks
- The introduction should be reviewed and reorganized in order to better clarify the research issues and the research goals; The story is difficult to follow. There are many gone-back in the introduction
- The state of the art should be revised. In this current form it is more a list of research work. The existing research proposals should be better analyzed with respect with the issues of the paper.
- I have the feeling that there are some confusion between, semantic, attribute, name, etc. There are not always used appropriate in the paper. For example, semantic represents the type of the features; the name is an attribute also. So the statement the “similarity between name the attibuts “ is not correct.
- The authors should better describe the novelty of the approach (by listing the new aspects they introduced). Better highlight the new insights of this research work. Can be added in the conclusion.
- Many English typos. Please revise.
Specific remarks
- What exactly are integrated pipelines? Which types of differences exist between integrated and professional pipelines: different models, dates, semantics, etc. These aspects are not clear in the paper and some description appears only in Section 5. It is difficult to understand why some choices in the proposed method without knowing first the issues due to the heterogeneities of data.
o Lines 43-46. This is not clear. What happens with the sources of data after matching? Fusion ? Links between homologous objects are kept to update both integrated and professional data?
- Stroke appears first line 60 without any definition before. Please define from the beginning what stroke is in this research work.
- Inconsistencies of attributes. Please give examples.
- Lines 69-71. Please be more specific. The paper (Xavier et al., 2016) can be cited.
- Lines 104-107. Please revise; the idea is not clear.
- The figure describing the proposed process should not be part of the introduction in my opinion. It should appear after the state of the art, because the goal of the research is to build based on the state of the art and to improve it. Moreover some steps illustrated in the figure are not easily found in the remainder of the paper. For example “Matching object detect candidate…” is not explained. I suppose that this step consists in defining the links between homologous strokes + some 1:1 links fusion to obtain n:m ). Please explain why the Geometry similarity and structural similarity criteria are applied in an iterative way. Lines 118: remove Section 1 from “the remainder of this paper…”
- Section 3. Give first the global idea for holistic stroke matching.
- Which is the cardinality of matched links for geometric and structural criteria?
- Equation 1 : how weighs are assigned? Justify why do you need to assess different weight to the three geometric measures; Justify why this choice for H(*) and A(*) where the measures are higher than the thresholds.
- Lines 226-230 and figure 3: difficult to match the text and the figure. Please revise.
- Equation 2 : what represents i and u?
- Section 4.2 : for the decomposition, please cite Voltz, 2006. Section 4.2 can be simplified.
- Section 5 : the thresholds and the weighs are empirically chosen. Nevertheless, the authors do not list the values, and how they are chosen (ex: by using histograms of values?). This should be absolutely be added in the paper.
- Since the data matching approach is applied step by step I suggest to evaluate (i.g. copute precision and recall) for each step. In this way it can be seen the relevance of each step.
- Please add discussions and statistics with respect with the cardinality of links.
- The conclusion should be completely reviewed. There is no discussion about the proposed approaches. Moreover, in lines 494-495 well known facts in the data matching literature are described; they should not be a conclusion of the paper.
References that can be studied:
1/ Samal, A., Seth, S. and Cueto, K. (2004). A feature-based approach to conflation of geospatial sources. IJGIS, vol. 18, n°5, p. 459-489 à spatial context
2/ Olteanu-Raimond, A., Mustière, S. and Ruas, A. 2015,’ Knowledge formalization for vector data matching using belief theory’, Journal of Spatial Information Science, (10). à a data matching approach applied on points and lines allowing managing inconsistencies and incompleteness;
3/ Xavier, E., Ariza-López, F. and Ureña-Cámara, M. 2016,’ A Survey of Measures and Methods for Matching Geospatial Vector Datasets’, ACM Computing Surveys, 49(2), pp.1-34. à A state of the art on data matching. Can also be used to better fix the vocabulary in the paper.
Reviewer 2 Report
The paper aims to develop a matching algorithm with improved matching accuracy, recall rate and running time.
Some statements need to be clarified before the paper is potentially published:
lines 177-178: The authors mention here "frequent N:M matching types" and continue "namely 1:1,..., N:M". Why "N:M" is doubled here?
Table 2:
For the integrated pipelines segment 661-662 the number of pipeline segment is stated "2". Reffering to the Figure 8(b) and matching type "1:1", is this number correct?
I would recommend to use plural in the name of the column, i.e. "The number of pipeline segments".
lines 389-435: There is only the text description of the algorithm, which is in similar way described in the previous sections. I would expect here more figures/numbers for each step demonstrating the principle of the algorithm.
line 441: The Equation (10) is mentioned, however no such Equation could be found in the text.
Reviewer 3 Report
This paper developed a holistic matching of pipeline strokes, and a partial matching algorithm to improve the network matching. I think this paper deals with an interesting topic and has the originality. However, I think that the completeness of the paper is so insufficient that it is deserved to be published after sufficient improvement and correction.
(1) The overall English writing is not good. There are many grammatical and typographical errors that need to be revised. For example, line 131 (two spatial targets [8, 10].In the matching,) is not properly spaced. Such a mistake in the spacing is in the entire article, so it needs to be corrected. There are also typographical errors in line 170 (Gong et al. [41] Gong et al. [41] semantic similarity presented), so please correct them. In addition, review the entire article and revise the typo.
(2) geo-information system is firstly used in line 32, and is subsequently used in the text as a GIS. However, there is no explanation for this abbreviation. In addition, a full name (geo-information system) is used again in line 123.
(3) In abstract, the authors said that ‘copes with spatial objects with topological changes, strong adaptability and extensibility to solve complex problems (growth, shortening, topology changes, etc.)’. However, I do not think there is any evidence for this comment in the text. In line 487, authors also said that ‘In addition, the holistic and partial match on the ability of dealing with complex problems (growth, shortening, topology changes, etc.) is better and the operation efficiency was high.’ Please explain in more detail how this study can solve complex problems.
(4) In this study, ‘pipeline stroke’ is mentioned and utilized as the basis of analysis. However, the lack of a description of the pipeline stroke makes it difficult for the reader to understand this study. A more detailed and basic explanation of the pipeline stroke is needed.
(5) There is a lack of explanation for some studies in the 'Related Work' chapter. If you simply list terms that readers do not know what they mean, readers cannot understand it. For example, a brief description for ‘mutual information function’ or ‘probability relaxation’ is needed.
(6) At line 177, the authors mention that: as well as frequent N: M matching types, namely 1: 1, 1: 0, 0: 1, 1: N,… Readers cannot understand exactly what it means, so a more detailed explanation is needed for this ratio.
(7) Line 207: A description of the ‘Hausdorff distance’ is required.
(8) Line 223: A description of the ‘spatial scene’ is required.
(9) In line 233, the authors state that: ‘According to strokes and the different relationships among them, a graph can be established.’ But I do not know where the graph is.
(10) In line 235, the authors said that: 'and every segment in this graph is noted as A, B, or C'. However, there are no figures for graph and A, B, and C that are mentioned in the text.
(11) Equation (2) should explain the difference between i, j and u, and the difference between n, m and k. There is no need to distinguish them if they are same. If they are different, the authors should explain the differences.
(12) Str and CanStr are mentioned in the following sentence: ‘For a certain stroke pair to match, they are denoted as set Str and set CanStr’. Readers can know that Str represents a stroke, but CanStr is not intuitive. Therefore, it is needed to explain what the CanStr is.
(13) Line 300: In the next sentence, I think that Figure 6 should be replaced with Figure 5. ‘In order solve the problem in Figure 6,’
(14) Line 302: In the next sentence, I think that SPMA-S should be replaced with SPMA-V. ‘In the stroke partial matching algorithm based on vertex decomposition (SPMA-S),’
(15) (Line 307) Similarly, the candidate matching pipeline stroke′of the stroke on another pipeline map is denoted as S′={ S′1 ,S′2 ,‥‥‥, S′k’,}, and the ordered vertex set is SN′= {p′1, p′2,‥ ‥‥,p′n}.: In this sentence, if k' is different from k, mention it in the text. Also, if n and n' are different numbers, then the last element of SN' should be p'n'.
(16) (Line 318) Similarly, the shortest distance is obtained for each node p ′j(where 1≤j≤m) of S′, and the Boolean spatial distance attribute value is obtained by using Equation (7). : If you use m in this sentence, you must also write m before it (not n'). Or if you write n' above, you should also write n’ here.
(17) (Line 336) In actual pipeline matching, stroke partial matching algorithm based on vertex decomposition (SPMA-V) is often used, but in the face of the growth, shortening, appearance and disappearance of pipelines, stroke partial matching algorithm based on vertex decomposition (SPMA-V) can play a better effect. : I do not understand this sentence. Perhaps SPMA-V seems to have been used incorrectly instead of SPMA-S.
(18) In Figure 6, I do not know the difference between the red arrow and the green arrow. Except for a few arrows, are not they connected to each other by red dot (vertax)?
(19) (1) Data preprocessing occurs suddenly on line 389, which is awkward in terms of the completeness of the paper and needs revision or explanation.
(20) Authors should increase the size of the text because readers cannot see any text in all the figures.
(21) In Table 2, 1: 1 is correct for the first matching type? 2 is correct for all the pipeline segment? Please check again.
(22) In Figure 10, the text is not visible. In Figure 10 (c), the strokes are indistinguishable and should be readable. In addition, there is no explanation for Figure 10 (a) in the text, so it should be supplemented. The difference between each figure should be explained in the caption. In addition, you need to explain the used method (SPMA-S or SPMA-V).
(23) Figures 11, 12, and 13 should be written differently in captions. 12 (b) and 13 (a) are not the same figure? If you have changed the threshold, explain this in the caption.
(24) The authors need to explain the advantages and originality of this method in the conclusion.